# Enhancement of the Thermal Performance of the Paraffin-Based Microcapsules Intended for Textile Applications

**DOI:** 10.3390/polym13071120

**Published:** 2021-04-01

**Authors:** Virginija Skurkyte-Papieviene, Ausra Abraitiene, Audrone Sankauskaite, Vitalija Rubeziene, Julija Baltusnikaite-Guzaitiene

**Affiliations:** 1Department of Textile Technologies, Center for Physical Sciences and Technology, 48485 Kaunas, Lithuania; ausra.abraitiene@ftmc.lt (A.A.); audrone.sankauskaite@ftmc.lt (A.S.); 2Department of Textiles Physical-Chemical Testing, Center for Physical Sciences and Technology, 48485 Kaunas, Lithuania; vitalija.rubeziene@ftmc.lt (V.R.); julija.baltusnikaite@ftmc.lt (J.B.-G.)

**Keywords:** paraffin PCM—melamine-formaldehyde microcapsules, outer shell, modification by MWCNTs and PEDOT: PSS, differential scanning calorimetry, dip coating, thermal conductivity and heat storage and release capacity, dynamic thermal behaviour

## Abstract

Phase changing materials (PCMs) microcapsules MPCM32D, consisting of a polymeric melamine-formaldehyde (MF) resin shell surrounding a paraffin core (melting point: 30–32 °C), have been modified by introducing thermally conductive additives on their outer shell surface. As additives, multiwall carbon nanotubes (MWCNTs) and poly (3,4-ethylenedioxyoxythiophene) poly (styrene sulphonate) (PEDOT: PSS) were used in different parts by weight (1 wt.%, 5 wt.%, and 10 wt.%). The main aim of this modification—to enhance the thermal performance of the microencapsulated PCMs intended for textile applications. The morphologic analysis of the newly formed coating of MWCNTs or PEDOT: PSS microcapsules shell was observed by SEM. The heat storage and release capacity were evaluated by changing microcapsules MPCM32D shell modification. In order to evaluate the influence of the modified MF outer shell on the thermal properties of paraffin PCM, a thermal conductivity coefficient (λ) of these unmodified and shell-modified microcapsules was also measured by the comparative method. Based on the identified optimal parameters of the thermal performance of the tested PCM microcapsules, a 3D warp-knitted spacer fabric from PET was treated with a composition containing 5 wt.% MWCNTs or 5 wt.% PEDOT: PSS shell-modified microcapsules MPCM32D and acrylic resin binder. To assess the dynamic thermal behaviour of the treated fabric samples, an IR heating source and IR camera were used. The fabric with 5 wt.% MWCNTs or 5 wt.% PEDOT: PSS in shell-modified paraffin microcapsules MPCM32D revealed much faster heating and significantly slower cooling compared to the fabric treated with the unmodified ones. The thermal conductivity of the investigated fabric samples with modified microcapsules MPCM32D has been improved in comparison to the fabric samples with unmodified ones. That confirms the positive influence of using thermally conductive enhancing additives for the heat transfer rate within the textile sample containing these modified paraffin PCM microcapsules.

## 1. Introduction

Nowadays the textile industry is more dynamic than ever before and successfully combines together the long-standing traditions with rapid technological progress. These trends are reflected in the increased focus on phase-changing materials (PCMs) known as “smart materials” which can improve thermal insulation and thermal comfort of textiles, through the thermo-regulating effect of the PCMs. The active temperature-regulating ability of PCMs nowadays is widely applied in thermal management of such areas as buildings, electronics, medicine, automotive, textiles. Considering application in textiles, PCMs integrated into various fibres and fabrics provide temporary warmth or coolness effect, thereby diminishing thermal discomfort [1,2,3,4,5]. Among available PCMs, organic PCMs with a phase change temperature range of 18 °C to 35 °C are the most appropriate for textile application [5]. Paraffinic hydrocarbons in the solid-liquid phase in the organic materials group are the most preferred and practical materials for the production of microencapsulated PCMs (MPCMs) [6]. The thermal, physical, chemical, and mechanical properties of MPCMs are heavily dependent on the raw materials and synthesis processes during microencapsulation [7]. The usually applied encapsulation material for paraffin PCM is physically and chemically stable [8] melamine-formaldehyde (MF) resin [9,10,11,12,13,14]. The polyurea-formaldehyde resin [15], polystyrene [16], and poly (methylmethacrylate) derivatives [17] were investigated as well.

MPCMs can be introduced into the textile materials by several main finishing methods: coating [18,19,20,21,22], impregnation and exhaust [23,24], and embedding into the fibre during the polymer matrix spinning [25,26,27]. The problem of paraffin PCM is its low thermal conductivity, for example, paraffin has a thermal conductivity of 0.22 W/(m∙K) [28] when compared with ˃3000 W/(m∙K) for multiwall carbon nanotubes (MWCNTs) [29]. Moreover, microencapsulated PCMs have a polymeric shell, which not only prevents the content from leaking but also resists heat transition at the same time [30]. It was observed that the improvement of MPCMs thermal properties and especially the effective thermal conductivity depends on microcapsules structure [31]. One of the possibilities is to improve the thermal conductivity of the core materials.

Carbon-based nanostructures (nanofibers, nanoplatelets, and graphene flakes), carbon nanotubes, both metallic (Ag, Al, C/Cu, and Cu) and metal oxide (Al_2_O_3_, CuO, MgO, and TiO_2_) nanoparticles and silver nanowires were investigated as the thermal conductivity promoters for materials of the PCMs [32,33,34]. However, it is difficult to ensure that the additives are uniformly distributed in the PCMs; moreover, the additives increase the weight of the PCMs and will decrease the latent heat storage capacity in general [35]. Another known opportunity is to enhance the thermal conductivity of MPCMs through the incorporation of nano-filler additives on polymeric shell materials. The thermal conductivity of the melamine urea-formaldehyde microcapsules containing paraffin PCM as a core has been significantly enhanced from 0.1944 W/(m∙K) to 1.0540 W/(m∙K) with a little influence to their enthalpy by a coating of 10 wt.% graphene sheets onto the polymeric shell [36]. The micro-PCM particles containing paraffin core with graphene/methanol modified melamine-formaldehyde hybrid shell have been successfully prepared and their microstructure and thermal performance were investigated as well [30,37]. Thermally conductive PCM paraffin-wax-embedded polymer microcapsules with graphene oxide (GO) platelet patched shell structure have been developed by researchers [38]. It was identified that the excellent thermal sensitivity of these microcapsules with GO platelet patched shell provides an efficient way to regulate thermal radiance according to the surrounding background, which made these microcapsule-embedded composites a promising material for active thermal camouflage and stealth applications. Electrically and thermally conductive paraffinic PCMs with melamine-formaldehyde shell microcapsules have been manufactured by coating with polypyrrole (PPy) and it is expected that these microcapsules will widen the application possibilities of PCMs in camouflage technology and electronic cooling [37].

Although different studies pertaining to the resistivity and conductivity properties of various fibre content fabrics modified by applying PPy [39] and PPy/carbon black composite [40] were analysed, a limited number of publications on the influence of the microencapsulated PCMs for the thermal resistance of textile materials were found. The researchers [41] found that the textile sample coated with polyethylene glycol (PEG) microcapsules during the testing with the Sweating Guarded-Hotplate apparatus has radiated 20% less heat than the untreated one.

This study is aimed to improve the thermal performance of paraffin PCM microcapsules for textile application by modifying their outer shell. For this purpose, the paraffin microcapsules MPCM32D with a transition temperature of 32.02 °C were modified using Layer-by-Layer (LbL) self-assembly technique to form multilayered thin coatings by electrostatic interaction among cationically charged MF resin shell and applied anionically charged thermal conductivity enhancing additives—MWCNTs or PEDOT: PSS. The presence of the layer of these additives on the outer shell of these modified PCM microcapsules was observed by a scanning electron microscope (SEM). The latent heat and thermal conductivity of the microcapsules MPCM32D modified with various concentrations of MWCNTs or PEDOT: PSS were measured, respectively, by the differential scanning calorimetry (DSC) technique and thermal conductivity determination device. The obtained results were evaluated by comparing them to the unmodified ones. In addition, a 3D warp-knitted spacer fabric from PET was dip-coated with shell-modified microcapsules MPCM32D that demonstrated optimal thermal characteristics, and its thermal performance—heat storage and release capacity, thermal conductivity, and dynamic thermal behaviour—was evaluated.

## 2. Materials and Methods

### 2.1. Materials

PCM microcapsules MPCM32D (composition: 17.4% melamine resin, 79.6% paraffin wax, water ≤ 3%) in a dry white powder were purchased from Microtek Laboratories Inc., Dayton, OH, USA. Multiwall carbon nanotubes NC7000 (average diameter: 9.5 × 10^−9^ m, average length: 1.5 μm) in form of 3 wt.% waterborne dispersion called Aquacyl AQ0302 were obtained from Nanocyl S.A., Sambreville, Belgium. Poly(3,4-ethylenedioxythiophene) poly(styrene sulfonate) (PEDOT: PSS) (1.3 wt.% dispersion in water, conductive grade), poly(diallyldimethylammonium chloride) (20 wt.% in water), sodium dodecylbenzenesulfonate (SDBS) and ethanol (95% denatured) were purchased from Sigma-Aldrich, Taufkirchen, Germany. Aqueous synthetic dispersion based on polyurethane (PU) Tubicoat MP and acrylic resin binder Itobinder PCM were obtained, respectively, from CHT Germany GmbH, Tübingen, DE and LJ Specialties Ltd., Chesterfield, Derbyshire UK.

#### 2.1.1. Modification of the Outer Shell of PCM Microcapsules with the Thermally Conductive Additives

The MF resin shell of the powdered PCM microcapsules MPCM32D was modified with thermally conductive additives MWCNTs and PEDOT: PSS, respectively, using a layer-by-layer self-assembly method [42]. First, 10 g of these PCM microcapsules and 1.2 g of cationic surfactant poly(diallyldimethylammonium chloride) (20 wt.% aqueous solution) were dispersed in 500 mL of ethanol and stirred at room temperature for 10 min using a high-speed disperser (1000 rpm). In parallel, the different mass fractions (1 wt.%, 5 wt.%, and 10 wt.%), respectively, 3.3 g, 16.5 g, and 33 g of MWCNTs 3% waterborne dispersion Aquacyl AQ0302, or 7.5 g, 37.5 g and 75g of 1.3 wt.% PEDOT: PSS aqueous dispersions and 0.1 g of the SDBS (anionic surfactants) were added to 500 mL of deionized water and stirred with a high-speed disperser (5000 rpm) for another 10 min. Finally, the prepared suspensions of ionized microcapsules MPCM32D and different parts of the thermal conductivity enhancing additives calculated by mass weight, respectively, were mixed under intensive shaking at 20 ± 5 °C for 60 min (considering the fact that positively charged outer shell of the microcapsules could attract the negatively charged conductive additives due to the electrostatic interaction). Then the powder of microcapsules was filtered and washed several times with 40 °C deionized water. The MWCNTs shell-modified PCM microcapsules MPCM32D were dried at room temperature. In case of modification with PEDOT: PSS, these microcapsules were dried in an oven for 10 min at 100 °C temperature for PEDOT: PSS film formation.

#### 2.1.2. Introduction of Shell-Modified PCM Microcapsules into Textile Materials

The unmodified and shell-modified PCM microcapsules analysed in this study were embedded in three-dimensional (3D) warp-knitted spacer fabric from PET and elastane, the technical data of which are presented in Table 1.

This knitted fabric was impregnated, respectively, with 5 wt.% MWCNTs or 5 wt.% PEDOT-PSS shell-modified PCM microcapsules MPCM32D, that demonstrated the best latent heat capacity and thermal conductivity characteristics, in composition with acrylic resin binder Itobinder PCM on a laboratory padder EVP-350 (Roaches International Ltd., West Yorkshire, UK) according to the recipe and process parameters presented in Table 2.

### 2.2. Methods of Investigation

#### 2.2.1. SEM Analysis

The surface morphology of the shell-modified PCM microcapsules MPCM32D and textile samples with these microcapsules was examined applying scanning electron microscopy (SEM) and using a Quanta 200 FEG device (FEI, Eindhoven, Netherlands) at low vacuum, 80 Pa, detector—LFD. All microscopic images were made under the same technical and technological conditions: electron beam heating voltage (10.00–30.00) kV, beam spot (3.0–5.0), magnification of 5000× and 10,000×, work distance (6.0–10.0) mm.

#### 2.2.2. DSC Analysis

The thermal and phase transition characteristics of the unmodified and shell-modified PCM microcapsules, as well as of the textile fabric containing these microcapsules were determined by standard method EN 16806-1 [43], using DSC module (DSC Q10, TA Instruments, New Castle, DE, USA) equipment under a nitrogenous atmosphere. The mass of test specimens was about 10 mg. A lid was pressed on the crucible to ensure good contact between the specimen and the bottom of the crucible. The micro-encapsulated PCM samples were dried at 60 °C for 24 h to remove all water. The samples underwent a heating-cooling-heating cycle from −20 °C to +60 °C at a heating and cooling rate of 5 °C/min. Based on the recorded second heating cycle, enthalpy of fusion in J/g was determined by measuring the area under the peak to the baseline constructed, using software of DSC Q10. In the same way, the enthalpy of crystallization in J/g was determined based on the recorded cooling cycle.

The transition temperatures—peak melting and peak crystallization temperatures—extrapolated onset and end melting as well as crystallization temperatures, were also determined based on respectively the recorded second heating or based on the recorded cooling cycle, as defined in EN ISO 11357-3 [44].

#### 2.2.3. Thermal Conductivity of PCM Microcapsules

Primarily, the unmodified and shell-modified PCM microcapsules have been incorporated into a polymeric matrix of polyurethane (PU) aqueous dispersion TUBICOAT MP and the homogeneous tablets in diameter of 55 mm and thickness of 3 mm were prepared. For this purpose, the 3 g of unmodified and shell-modified microcapsules MPCM32D and 3 g of TUBICOAT MP were intensively stirred until rigid consistency paste was achieved. Finally, the prepared mass was inserted into the round metal form and dried at 50 °C for about 8 h. For easier removal of the content from the form, a cellulose film was placed at the bottom. In the beginning, this film was overheated at a higher temperature than the drying temperature of the tablets to prevent the film from shrinking. Next, the tablets were stored in a desiccator for 24 h. The images of the formed tablets are shown in Figure 1.

The thermal conductivity measurements of the shell-modified microcapsules MPCM32D in a PU matrix were performed using an original thermal conductivity determination device [45] where the known thermal conductivity material as a reference was compared with the testing one. Next, these shell-modified PCM microcapsules with the best thermal performance were introduced into the textile.

#### 2.2.4. Thermal Conductivity of Textile Materials Containing PCM Microcapsules

The thermal conductivity of the developed 3D warp-knitted spacer PET fabric containing microcapsules MPCM32D was evaluated based on the measurements of the reciprocal parameter—thermal resistance (*R_ct_*). This parameter was measured under steady-state conditions using sweating guarded—hotplate M259B (SDL Atlas, Rock Hill, SC, USA) according to the standard method EN ISO 11092 [46]. The principle of this method is that the sample to be tested is placed on an electrically heated plate with conditioned air ducted flow across and parallel to its upper surface. For the determination of the thermal resistance, the heat flux through the test specimen is measured after steady-state conditions have been reached. Thermal resistance was calculated by the formula [46]:(1)$$R_{ct}=\;\frac{\left(T_{m}-T_{a}\right)\;A\;}{H-\Delta H_{c}}-R_{ct0}\;(m^{2}\;K/W),$$
where *T_m_* = 35 °C (temperature of the measuring unit); *T_a_* = 20 °C (air temperature in the test enclosure); *H*—heating power supplied to the measuring unit, in watts; *A*—area of the measuring unit in square meters; *R_ct0_*—the apparatus constant, in square meters Kelvin per Watt; Δ*H_c_* = 0 (correction term for heating power).

The coefficient of thermal conductivity λ, W/(m·K) was calculated according to the formula [47]:*λ = D/R_ct_* (W/(m·K)),(2)
where: *D*—thickness of the sample (m); *R_ct_*—thermal resistance (m^2^ K/W).

#### 2.2.5. Evaluation of Dynamic Thermal Behaviour of Textile Materials Containing PCM Microcapsules

The dynamic thermal behaviour of 3D warp-knitted spacer PET fabric dip-coated with unmodified and shell-modified microcapsules MPCM32D were analysed by using a thermal camera (spectral range: *λ* = 7.5 ÷ 13 µm) InfraCAM (FLIR SYSTEMS AB, Täby, Sweden) and an IR emitting lamp (250 W, 240 V, *λ* = 500–3000 nm, Ø = 125 mm) as the heat source (Figure 2). After 4 min as the IR lamp was switched on, the hottest location on the testing table surface was detected by using the InfraCAM. Thereafter, a polystyrene foam plate with a flatly laid spacer fabric was centred on the marked hottest location. The samples were heated for 4 min with an IR lamp placed about 50 cm above the knitted fabric to achieve a temperature of 4–5 °C higher than the melting point (32.02 °C) of paraffin PCM. Then the lamp was switched off, the knitted fabric was allowed to cool for another 4 min. The fabric samples were observed by the thermal camera every 15 s during the entire 8 min of the test. Five temperature measurements for each sample were performed and the imaginary temperature value was calculated. The variation coefficient was <5%.

## 3. Results and Discussion

### 3.1. Surface Morphology of The Shell-Modified PCM Microcapsules

Before the microscopic analysis, the visual observation of the shell-modified microcapsules MPCM32D was performed. It was observed that after the modification with the thermally conductive additives MWCNTs or PEDOT: PSS, the initial white colour of these paraffin PCM microcapsules has changed to black or blue, respectively. The colour and its depth of these shell-modified microcapsules depended on the nature of the used additive and its concentration in the modification process (see Figure 3). It is especially expected that textile with the embedded blue coloured PEDOT: PSS shell-modified PCM microcapsules could be promising for leisure and protective clothing not only because of the enhanced thermal performance but also due to the expansion of colouristic possibilities to avoid the black colour.

In order to estimate the surface morphology and the form of the shell-modified microcapsules MPCM32D, the SEM analysis was made. The micrographs of the modified, as well as unmodified microcapsules detected by SEM, are shown in Figure 4. A microgram (Figure 4a) shows an optical view of the unmodified MPCM32D microcapsules—they have a smooth and compact surface, and have a regular sphere shape. The images of shell-modified microcapsules presented in Figure 4 clearly show the newly formed coating of MWCNTs (Figure 4b,c) and PEDOT: PSS (Figure 4d,e), and this confirms the presence of these conductive additives on the outer shell of the PCM microcapsules.

### 3.2. Heat Storage and Release Capacity of the Shell-Modified PCM Microcapsules

The results of DSC analysis for unmodified and shell-modified microcapsules MPCM32D are presented in Figure 5 as the DSC curves—thermograms. The data extracted from DSC curves for heating and cooling processes of all tested paraffin PCM microcapsules are summarized in Table 3. The initial, unmodified microcapsules MPCM32D consisting of an MF resin shell surrounding a paraffin core, demonstrated the peak melting temperature of 32.02 °C (Figure 5a). This result falls within the melting temperature range (30–32 °C) specified by the manufacturer and corresponds to a typical paraffin two-peak DSC thermogram. The sharp main peak of the curve represents the solid-liquid phase change of the tested encapsulated paraffin PCM and the minor peak at the left side of the main peak corresponds to the solid-solid phase transition of paraffin. In this case, for unmodified microcapsules MPCM32D, the enthalpy of fusion (ΔH_f_) of the solid-solid and the solid-liquid transition is 10.4 J/g and 106.3 J/g, respectively. After the modification, these two-phase change peaks remain (see Figure 5b,c) and this implies that modified microcapsules maintain good phase change behaviour. As enthalpies of fusion (ΔH_f_) and crystallization (Δ*H_c_*) during the solid-solid phase changing stage are significantly lower (for unmodified and shell-modified microcapsules) if compared to the solid-liquid stage, therefore for the further analysis only enthalpies of the solid-liquid stage were considered. The data calculated from DSC curves for shell-modified microcapsules MPCM32D with MWCNTs and PEDOT: PSS in different mass fractions (1 wt.%, 5 wt.%, and 10 wt.%) are presented in Table 3.

The melting peaks of the modified microcapsules MPCM32D with both thermally conductive additives, MWCNTs or PEDOT: PSS, shifted to a slightly higher temperature (max. by ~2 °C) and their crystallization peaks moved to a slightly lower temperature (max. by ~3 °C). Hence, it may be concluded that for the modified microcapsules MPCM32D, in both cases—using as additives either MWCNTs or PEDOT: PSS—phase transition temperatures are quite similar to those of the unmodified. The variations between these temperature values are rather small and are within the measurement uncertainty limits. It demonstrates that the incorporation of used outer shell modifiers does not affect the structure of microcapsules MPCM32D paraffin core. Moreover, some researchers [48,49,50] have proved that there was no chemical reaction between the paraffin and the CNTs or graphite in the preparation of their composites.

As seen in Table 3 and Figure 6, the modification of the MF resin outer shell of these microcapsules has a certain influence on their heat storage and release capacity.

These properties are the most important part of PCM application and define their thermoregulatory activity. The enthalpy of fusion (ΔH_f_) of the 1 wt.% and 5 wt.% MWCNTs, as well as PEDOT: PSS, shell-modified paraffin microcapsules MPCM32D was found to be approximately the same as that of the unmodified ones. Only an increase in the amount of these thermally conductive additives until 10 wt.%, significantly reduced the heat storage capacity of the investigated PCM microcapsules. If compared with the unmodified ones, ΔH_f_ decreased 15.6% for MWCNTs and 12.3% in the case of PEDOT: PSS. A similar situation was observed with the enthalpy of crystallization (Δ*H_c_*). This can be explained by the fact, that the addition of 10 wt.% or more of this thermal conductivity and heat transfer enhancing additives has quite considerably decreased the relative amount of paraffin PCM (or relative fraction of this PCM) in the microcapsules. Therefore, on the basis of this analysis, it could be stated, that the shell-modified microcapsules MPCM32D with the mass fraction of additives—MWCNTs or PEDOT: PSS—at about 5 wt.%, may be optimal for intended textile applications. The thermograms of 5 wt.% MWCNTs and 5 wt.% PEDOT: PSS shell-modified microcapsules MPCM32D, respectively, are shown in Figure 5b,c, and a thermogram of these three curves is summarized in Figure 5d.

### 3.3. Influence of Conductive Additives on the Thermal Conductivity of Modified PCM Microcapsules

In order to evaluate the influence of the used modifiers—MWCNTs or PEDOT: PSS—for the heat transfer efficiency of microcapsules MPCM32D, the thermal conductivity of the modified and unmodified microcapsules was measured. The thermal conductivity is considered as an important contributing factor to the total thermoregulatory capability of PCMs microcapsules, as it may delay or promote the thermal response to the heat storage and release [49,51]. With an aim to determine the thermal conductivity, the microcapsules MPCM32D have been incorporated into a polymeric matrix of polyurethane (PU) and obtained samples were tested. Therefore, the results presented in Figure 7 are not directly related to the particular tested microcapsules, but to the composites containing unmodified microcapsules MPCM32D and modified ones with different mass fractions (1 wt.%, 5 wt.%, and 10 wt.%) of MWCNTs or PEDOT: PSS. Thus, these results (Figure 7) allow for comparing the thermal conductivity of PCM microcapsules modified with different quantities of thermally conductive additives and to evaluate the effect. The thermal conductivity coefficient λ (W/(m K)) of each prepared microcapsules sample in the PU matrix is expressed in Figure 7. The effect of different mass fractions of both modifiers (MWCNTs and PEDOT: PSS) on the thermal conductivity of tested samples is also provided. It is evident that the thermal conductivity of the tested samples showed a linear increase with the increment of MWCNTs or PEDOT: PSS mass fraction by modifying the outer shell of the paraffin PCM microcapsules. Furthermore, it is obvious that the nature of the used additives has influenced these properties. The samples of microcapsules MPCM32D modified with MWCNTs demonstrated a higher enhancement of thermal conductivity compared to the ones modified with PEDOT: PSS. This is due to the higher internal conductivity of MWCNTs. After the comparison of the thermal conductivity values of the samples containing unmodified microcapsules MPCM32D (λ = 0.105 W/(m∙K)) with the samples modified with 10 wt.% MWCNTs (*λ* = 0.264 W/(m∙K) a remarkable improvement (2.5 times or 151%) is seen. Meanwhile, the thermal conductivity of microcapsules samples which were shell-modified with 10 wt.% PEDOT: PSS revealed less improvement (1.8 times or 79%).

### 3.4. Thermal Performance of Knitted Fabrics Containing Modified PCM Microcapsules

Based on the gained results of the phase change characteristics and the thermal conductivity of the shell-modified microcapsules MPCM32D, it was determined that mass fraction of MWCNTs or PEDOT: PSS, not exceeding 5 wt.%, may be the optimal one for textile applications, as the higher amount of these conductive additives reduces the heat storage and release capacity of paraffin PCM. Consequently, the 3D warp-knitted spacer PET fabric (Table 1) was dip-coated with the Itobinder PCM and 5 wt.% MWCNTs or 5 wt.% PEDOT: PSS shell-modified microcapsules MPCM32D according to the recipe and parameters presented in Table 2. For comparison, samples of knitted fabrics—untreated, dip-coated with unmodified microcapsules MPCM32D, and treated only with Itobinder PCM were used and their codes are given in Table 4.

For the evaluation of the thermal performance of these samples, three testing methods were applied—DSC analysis (except Samples 1 and 2), determination of thermal resistance (*R_ct_*) under steady-state conditions using sweating guarded-hotplate, and monitoring of dynamic thermal behaviour, during the temperature changes, based on infrared thermography. The DSC results of Samples 4 and 5 presented in Table 5, revealed that the fabric with 5 wt.% MWCNTs or 5 wt.% PEDOT: PSS shell-modified microcapsules MPCM32D feature quite similar enthalpies of fusion (ΔH_f_) and crystallization (Δ*H_c_*) when compared to the fabric containing the unmodified microcapsules (Sample 3). Hence, it can be concluded that the heat storage and release capacities for 3D warp-knitted spacer PET fabric impregnated with 5 wt.% MWCNTs or PEDOT: PSS shell-modified paraffin microcapsules MPCM32D remain the same as in the case of unmodified microcapsules.

The DSC analysis allows obtaining the basic thermal properties of the textile samples treated with PCM microcapsules, mostly related to their thermoregulatory capacity. However, to assess the thermal performance of the tested textile samples considering the thermal response rate, two more methods were used. To assess the influence of the microcapsules modification on the thermal conductivity of textile samples, the thermal resistance *R_ct_,* (m^2^∙K/W) was measured. Based on the obtained results the thermal conductivity coefficient *λ*, (W/(m∙K)) was calculated. The received values are presented in Table 4. The structure of the knitted fabric and the layer of polymeric binder used for the incorporation of PCM microcapsules have influenced the heat transportation to these microcapsules and the temperature values. This might be because of the differences in the thermal conductivity of individual components of the tested material and their distribution. The tested textile samples consist of PET/elastane yarns, an acrylic resin binder, and PCM microcapsules, therefore the thermal conductivity was determined separately for knitted fabric: untreated (initial) (Sample 1), dip-coated only with Itofinish PCM (Sample 2), dip-coated with unmodified (Sample 3) and with 5 wt.% PEDOT: PSS and MWCNTs shell-modified (Samples 4 and 5) microcapsules MPCM32D, respectively. As it is seen from the results presented in Table 4, the thermal conductivity of Sample 2 is slightly lower than that of Sample 1. It suggests the conclusion that the acrylic resin binder is a barrier, so it can delay heat access and its emission. Furthermore, the thermal conductivity of the fabric sample with unmodified microcapsules (Sample 3) decreased even more, as one more barrier—MF resin shell of microcapsules—appeared. However, after modification of the outer shell of the microcapsules MPCM32D with 5 wt.% of used add-on’s, the thermal conductivity of investigated knitted fabric has improved in comparison to the fabric samples with unmodified ones: ~12% in case of MWCNTs modifier and ~ 9% for PEDOT: PSS. That confirms the positive influence of the used thermal conductivity enhancing additives for the heat transfer rate within the textile sample containing these modified paraffin PCM microcapsules.

To obtain supplementary information regarding the heat transfer capabilities of modified PCM microcapsules the dynamic thermal behaviour of fabric samples containing a composition of the unmodified and shell-modified microcapsules MPCM32D and acrylic resin binder Itofinish PCM, during the temperature changes was investigated. The results of the analysis (Figure 8) showed that Samples 4 and 5 containing shell-modified PCM microcapsules heated up faster than the samples containing unmodified (Sample 3) microcapsules whose behaviour was practically indistinguishable from that of untreated fabric sample’s (Sample 1). The results of these measurements showed that the formed coating of the MWCNTs and PEDOT: PSS on the outer shell of paraffin microcapsule MPCM32D accelerated their melting process and increased the heat transfer rate of the material, which ensured a rapid thermal response to the ambient temperature. The results of these measurements revealed that coating of the outer shell of the paraffin microcapsules MPCM32D with the thermal conductivity enhancing additives MWCNTs and PEDOT: PSS accelerates their melting process and increases the heat transfer rate of the fabric, which ensures a rapid thermal response to the ambient temperature.

## 4. Conclusions

Shell-modification of PCM microcapsules MPCM32D with MWCNTs and PEDOT: PSS additives resulted in enhancement of their thermal conductivity. The SEM micrographs have shown a clearly visible change in surface topography of modified microcapsules due to the presence of modifiers used on their outer shell surface. This fact, as well as essentially unaltered phase transition temperatures of modified microcapsules, confirms that both conductive additives—either MWCNTs or PEDOT: PSS—are adhered only to the outer shell of PCM microcapsules.

It was determined the optimal amount of conductive additive in the modified microcapsules, which is about 5 wt.% for both modifiers. The higher amount of applied additives (10 wt.% and more) despite significant improvement of thermal conductivity, rather reduces the heat storage and release capacity of investigated PCM microcapsules.

The investigation of the thermal performance of textile samples confirmed the positive influence of these paraffinic microcapsules’ shell-modification. For samples treated with shell-modified microcapsules MPCM32D in comparison with samples treated with unmodified ones: the thermal conductivity increased ~12% in the case of MWCNTs modifier, and ~9% in the case of PEDOT: PSS; the heat transfer rate through the material increased, ensuring more rapid thermal response to the ambient temperature. DSC analysis results proved that these modified microcapsules incorporated into textile maintain good heat storage and release capacity similar to unmodified microcapsules MPCM32D. Besides, it should be noted that investigations on the durability of fabrics treated with composites of modified microcapsules and acrylic resin binder will be the subject of our subsequent study.

## Figures and Tables

**Figure 1 polymers-13-01120-f001:**
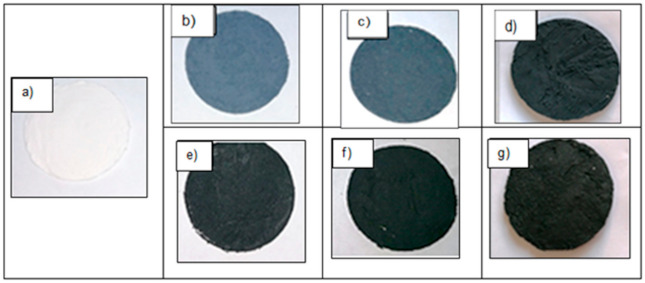
Images of unmodified microcapsules (**a**) and shell-modified MPCM32D microcapsules with different mass fraction: (**b**) poly (3,4-ethylenedioxyoxythiophene) poly (styrene sulphonate) (PEDOT: PSS) 1 wt.%, (**c**) PEDOT: PSS 5 wt.%, (**d**) PEDOT: PSS 10 wt.%, (**e**) multiwall carbon nanotubes (MWCNTs) 1 wt.%, (**f**) MWCNTs 5 wt.%, (**g**) MWCNTs 10 wt.% in a PU matrix.

**Figure 2 polymers-13-01120-f002:**
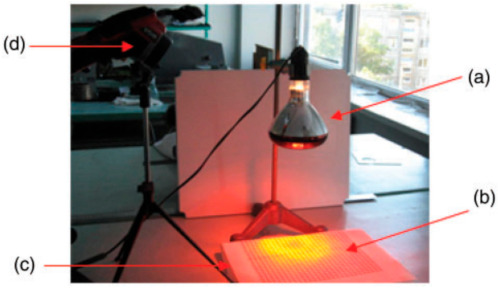
Stand for the analysis of fabric dynamic thermal behaviour: (**a**) infrared lamp; (**b**) 3D warp-knitted spacer PET fabric; (**c**) polystyrene foam plate; (**d**) thermal camera.

**Figure 3 polymers-13-01120-f003:**
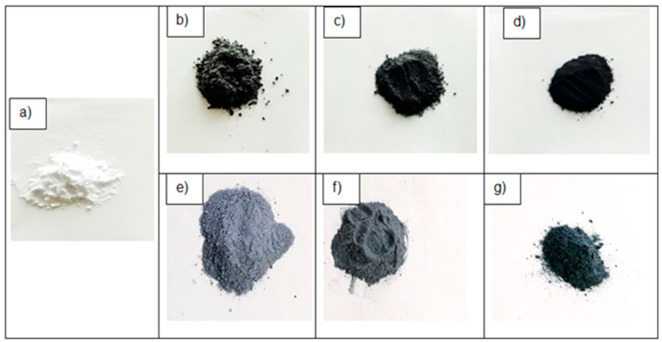
Pictures of microcapsules MPCM32D in powder form: unmodified (**a**) and shell-modified with different mass fraction: (**b**) MWCNTs 1 wt.%, (**c**) MWCNTs 5 wt.%, (**d**) MWCNTs 10 wt.%, (**e**) PEDOT: PSS 1 wt.%, (**f**) PEDOT: PSS 5 wt.%, (**g**) PEDOT: PSS 10 wt.%.

**Figure 4 polymers-13-01120-f004:**
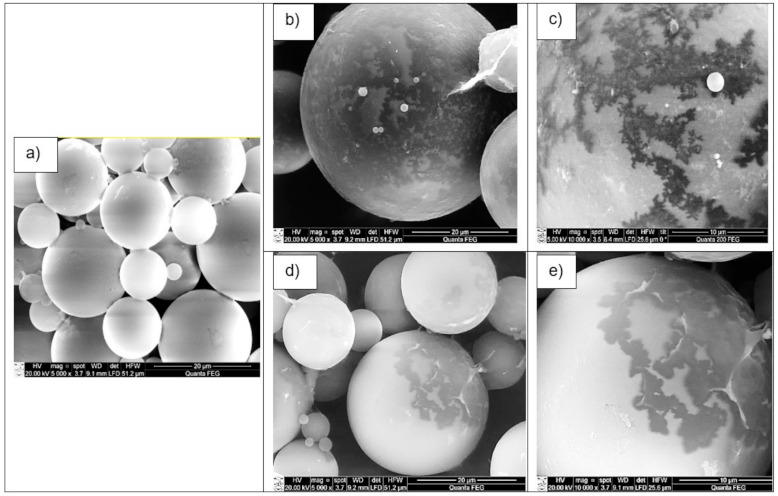
SEM micrograms of microcapsules MPCM32D: (**a**) unmodified, magnification 5000×; (**b**) with 5 wt.% MWCNTs shell-modified, magnification 5000×; (**c**) with 5 wt.% MWCNTs shell-modified, magnification 10,000×; (**d**) with 5 wt.% PEDOT: PSS shell-modified, magnification 5000×; (**e**) with 5 wt.% PEDOT: PSS shell-modified, magnification 10,000×.

**Figure 5 polymers-13-01120-f005:**
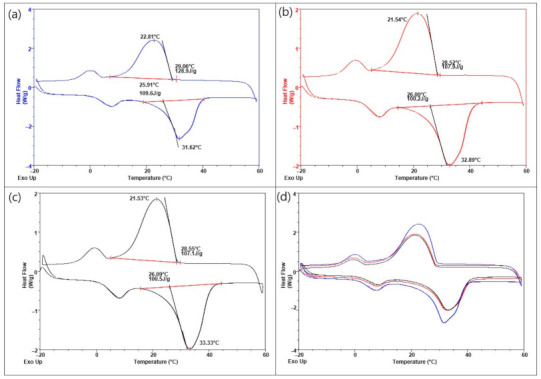
DSC thermograms of the microcapsules MPCM32D: unmodified (**a**), with 5 wt.% MWCNTs (**b**) and 5 wt.% PEDOT: PSS (**c**) shell-modified, and their summed thermogram (**d**).

**Figure 6 polymers-13-01120-f006:**
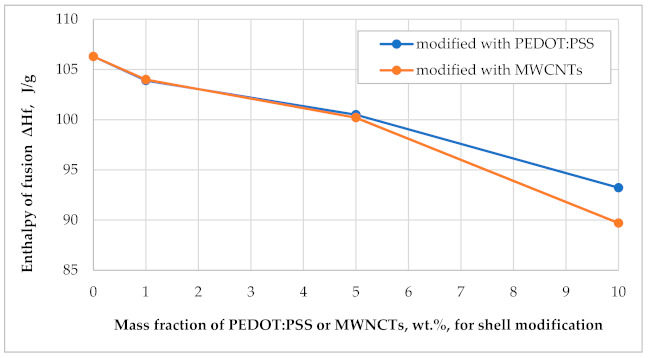
The influence of shell modifiers PEDOT: PSS and MWCNTs on fusion enthalpy of paraffin phase changing materials (PCMs) microcapsules.

**Figure 7 polymers-13-01120-f007:**
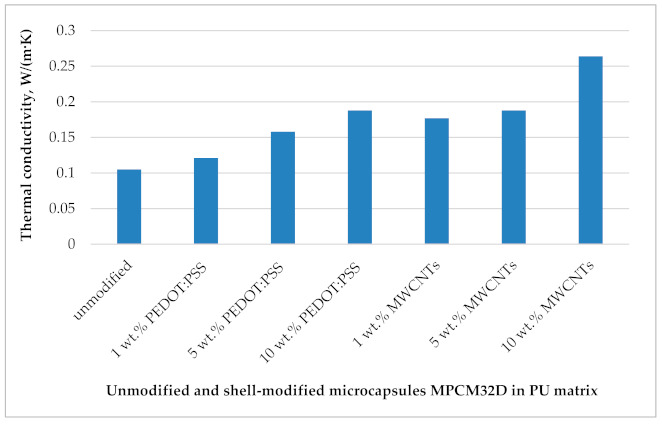
Thermal conductivity of unmodified and shell-modified PCM microcapsules MPCM32D in a polyurethane matrix.

**Figure 8 polymers-13-01120-f008:**
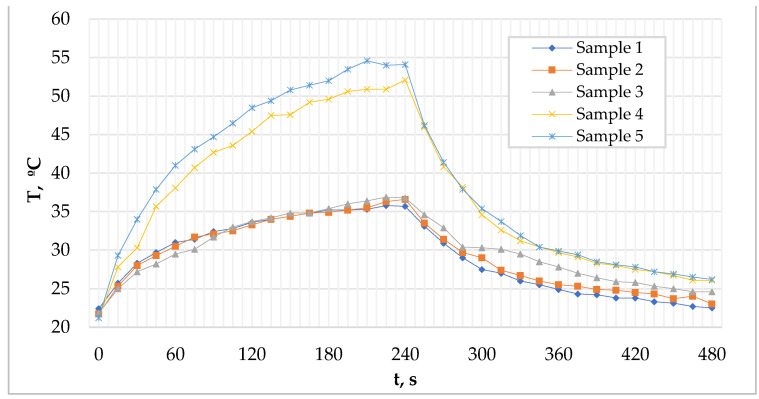
Comparison of dynamic thermal behaviour of the fabric samples containing unmodified and shell-modified PCM microcapsules during the temperature changes.

**Table 1 polymers-13-01120-t001:** Characteristics of 3D warp-knitted spacer fabric.

3D Warp-Knitted Spacer Fabric View—Warp-Wise	Type of Yarn and Linear Density, Tex	Content of Yarn, %	Mass per Unit Area, g/m^2^	Thickness, mm
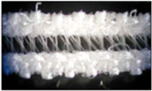	PET textured, 11.0	70	358	2.9
	PET monofilament, 5.6	20		
	Elastane, 7.8	10		

**Table 2 polymers-13-01120-t002:** Parameters of dip coating and drying-curing processes.

No.	Process	Auxiliaries		Parameters
		Microcapsules MPCM32D, g/L	Itobinder PCM, g/L	
1	Dip coating with microcapsules MPCM32D: Unmodified 5 wt.% MWCNTs shell-modified 5 wt.% PEDOT: PSS shell-modified	66	200	Wet pick-up: 80% Nip rolls: 2 bar
2	Drying—curing			Temperature: 120 °C
				Time: 6–8 min

**Table 3 polymers-13-01120-t003:** DSC results of the unmodified and shell-modified microcapsules MPCM32D.

Microcapsules MPCM32D	Melting				Crystallization			
	Peak Melting Temperature, Tp.m. °C	Extrapolated Onset Melting Temperature, Tei.m*.°*C	Extrapolated End Melting Temperature, Tef.m. °C	Enthalpy of Fusion, ΔH_f_*,* J/g	Peak Crystallization Temperature, Tp.c*.°*C	Extrapolated Onset Crystallization Temperature, Tei.c. °C	Extrapolated End Crystallization Temperature, Tef.c*. °* C	Enthalpy of Crystallization, Δ*Hc*, J/g
Unmodified (initial)	32.02	13.19	40.09	106.3	21.44	30.61	5.49	113.0
Modified with PEDOT:PSS:								
1 wt.%	32.73	15.68	44.9	103.9	21.56	30.13	5.01	112.9
5 wt.%	33.33	15.56	45.06	100.5	21.53	30.25	3.95	108.5
10 wt.%	33.49	15.4	44.9	93.22	21.27	29.8	4.09	103.1
Modified with MWCNTs:								
1 wt.%	33.85	15.0	42.2	104.0	22.9	30.3	5.31	106.1
5 wt.%	32.89	14.49	45.42	100.2	21.54	29.42	4.42	108.7
10 wt.%	33.35	14.85	43.05	89.77	22.21	31.3	5.25	96.4

**Table 4 polymers-13-01120-t004:** Thermal properties of 3D warp-knitted spacer PET fabric untreated and dip-coated with microcapsules MPCM32D.

Sample Code	Method of Textile Sample Treatment	Thermal Resistance, *R_ct_* (m^2^∙K/W)	Thermal Conductivity Coefficient, *λ* (W/(m·K))
1	Untreated (initial)	0.068	0.043
2	Dip-coated with Itofinish PCM	0.069	0.042
3	Dip-coated with Itofinish PCM and unmodified microcapsules MPCM32D	0.073	0.039
4	Dip-coated with Itofinish PCM and 5 wt.% PEDOT:PSS shell-modified microcapsules MPCM32D	0.061	0.047
5	Dip-coated with Itofinish PCM and 5 wt.% MWCNTs shell-modified microcapsules MPCM32D	0.060	0.048

* The thickness *D* (m) of all samples is the same, *D* = 0.0029 m.

**Table 5 polymers-13-01120-t005:** DSC analysis results of 3D warp-knitted PET fabric with unmodified and shell-modified microcapsules MPCM32D.

Sample Code	Knitted Fabric with Microcapsules MPCM32D	Melting		Crystallization	
		Peak Melting Temperature, Tp.m, °C	Enthalpy of Fusion, ΔH_f_, J/g	Peak Crystallization Temperature, Tp.c, °C	Enthalpy of Crystallization, Δ*H_c_*, J/g
3	Unmodified	31.66	21.78	24.78	24.64
4	5 wt.% PEDOT:PSS shell-modified	31.42	22.71	24.28	23.19
5	5 wt.% MWCNTs shell-modified	31.42	24.22	24.65	26.31

## Data Availability

The data presented in this study are available on request from the corresponding author.

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
