# Peer review of "Enhancement of the Thermal Performance of the Paraffin-Based Microcapsules Intended for Textile Applications"

_polymers, 2021, doi:10.3390/polym13071120_

Round 1

Reviewer 1 Report

Reviewers' comments:

Manuscript Number: polymers-1141461

Full Title: Enhancement of the Thermal Performance of the Paraffin-Based Microcapsules Intended for Textile Applications.

Comments: 

The manuscript reported on Enhancement of the Thermal Performance of the Paraffin-Based Microcapsules Intended for Textile Applications. The manuscript needs a detailed editing. It cannot be recommended for publication in the present form. I hope the following points would be helpful for the authors.

- In the Abstract - it's too long, the authors need to improve with more specific short results and conclusions.

- The introduction section should be improved.

- 2.2.2. DSC analysis - should be improve.

- Please provides the references for all equations and formula.

- Figure 5. Not clear maker clear.

- Figure 6. Not clear maker clear.

- Several faults: are added or missing spaces between words: see manuscript file.

- In part SEM how the energy of the accelerator beam used?

- Conclusion should be concise.

- References: make all references in same format for volume number, page numbers and journal name, because it is difficult to searching and reading.

- Some sentences need reconstruction and the level of English should be improved.

So that I recommended this manuscript to major revision and for future process.

Author Response

Point 1: - In the Abstract - it's too long, the authors need to improve with more specific short results and conclusions.

 Response 1: The Abstract was shortened. The new version is presented in revised manuscript attached.

Point 2: - The introduction section should be improved.

Response 2: The corrected version is presented in revised manuscript attached.

Point 3: - 2.2.2. DSC analysis - should be improve.

Response 3: DSC analysis was supplemented.

Point 4: - Please provides the references for all equations and formula.

Response 4: References are provided for both equations.

Point 5: - Figure 5. Not clear maker clear.

Response 5: Figure 5 is improved (please, see the revised manuscript).

Point 6: - Figure 6. Not clear maker clear.

Response 6: Figure 6 is improved (please, see the revised manuscript).

Point 7: - Several faults: are added or missing spaces between words: see manuscript file.

Response 7: The text was checked ones more and faults corrected.

Point 8: - In part SEM how the energy of the accelerator beam used?

Response 8: The SEM methodology has been corrected.

Point 9: - Conclusion should be concise.

Response 9: The recommendation is fulfilled – conclusions were revised.

Point 10: - References: make all references in same format for volume number, page numbers and journal name, because it is difficult to searching and reading.

Response 10: The recommendation is fulfilled – references were uniformed.

Point 11: - Some sentences need reconstruction and the level of English should be improved.

Response 11: The text has been corrected.

Reviewer 2 Report

Please find the comments below:

1) In the coating methods and thermal conductivity measurements on textiles important citation missing R. Villanueva et al., Materials Research Express 6, 016307 (2019)

2) Thermal conductivity vs temperature needed in comparison with bare textiles

3) Need wash cycle testing for durability 

4) Figure captions has with units in brackets needs to be improved

Author Response

Point 1: In the coating methods and thermal conductivity measurements on textiles important citation missing R. Villanueva et al., Materials Research Express 6, 016307 (2019)

Response 1: The recommendation is fulfilled – citation was added.

Point 2: Thermal conductivity vs temperature needed in comparison with bare textiles

Response 2: In manuscript the untreated textile sample is marked as Sample 1. The comparison of thermal conductivity of untreated and treated textile samples are given in text after Table 5. Besides, the thermal conductivity was calculated from thermal resistance which was obtained under steady-state conditions with a constant temperature of measuring unit (35 °C) – experimental part 2.2.4c.

Subjecting to the note, the text (last paragraph of Results and Discussion part) was supplemented with comparison of dynamic thermal behaviors of untreated and treated samples of fabrics.

Point 3: Need wash cycle testing for durability 

Response 3: We appreciate this observation and kindly inform that investigations on durability of directly binded modified microcapsules to textiles surface will be the subject of our subsequent study.

Point 4: Figure captions has with units in brackets needs to be improved

Response 4: The recommendation is fulfilled –captions of figures were rewritten.

Round 2

Reviewer 1 Report

Reviewers' comments:

The authors revised the manuscript according to the reviewers' comments.

So that I recommended this manuscript accept for publication in Polymers.

Reviewer 2 Report

Accept at its current form.